# Health Information Technology Use and Patient Safety: Study of Pharmacists in Nebraska

**DOI:** 10.3390/pharmacy7010007

**Published:** 2019-01-10

**Authors:** Kimberly A. Galt, Kevin T. Fuji, Ted K. Kaufman, Shweta R. Shah

**Affiliations:** 1Center for Health Services Research and Patient Safety, Creighton University, Omaha, NE 68178, USA; kfuji@creighton.edu (K.T.F.); TedKaufman@creighton.edu (T.K.K.); srshah6@wisc.edu (S.R.S.); 2Department of Pharmacy Sciences, Creighton University School of Pharmacy and Health Professions, Omaha, NE 68178, USA; 3Department of Pharmacy Practice, Creighton University School of Pharmacy and Health Professions, Omaha, NE 68178, USA; 4School of Pharmacy, University of Wisconsin, Madison, WI 53705, USA

**Keywords:** pharmacy, health information technology, patient safety, medication safety, error reporting, practice culture, mixed methods

## Abstract

This study aimed to describe the impact of 13 different health information technologies (HITs) on patient safety across pharmacy practice settings from the viewpoint of the working pharmacist. A cross-sectional mixed methods survey of all licensed practicing pharmacists in 2008 in Nebraska (*n* = 2195) was developed, pilot-tested and IRB approved. One-fourth responded (24.4%). A database of pharmacists’ responses to closed-ended quantitative questions and in vivo qualitative responses to open-ended questions was built. Qualitative data was coded and thematically analyzed, transformed to quantitative data and descriptive and relational statistics performed. One-third were involved in an error of any kind in the six months preceding the survey, and half observed an error or “near miss”. Most errors or near misses were attributed to workload. When asked specifically about the 13 HITs, these participants reported 3252 observations about the types of errors that were associated with each. These were reports about either error types reduced or eliminated by integration of HIT (*n* = 1908) or occurring in association with a specific technology’s use (*n* = 1344). Integration of HIT into pharmacy practice also introduced new error types such as excessive alert programming in the pharmacy computer systems clinical information support causing pharmacists to experience alert fatigue and ignore warnings or bar code scanners mismatching NDC codes of products resulting in wrong drug product identification. Continued vigilance is essential to identifying patient safety issues and implementing safety strategies specific to each HIT.

## 1. Introduction

Risks of harm and injury to patients are associated with health information technology (HIT) implementation and use, introducing new safety challenges into the healthcare system [1]. Error reduction is a primary goal for introducing HIT, but with these new risks emerge unintended consequences which negatively impact patient safety and quality [2,3,4,5]. Integration of HIT creates new errors due to improper design, development, use and implementation [1]. Twenty years have passed since a series of sentinel reports describing the major risks of harm and injury with health care were released (*To Err is Human*; *Crossing the Quality Chasm*; and *Preventing Medication Errors*) [6,7,8]. These reports highlight the central role of HIT to risk reduction and identify pharmacists and pharmacy services as central to the incorporation of new technologies into health care delivery, and implementation of prevention and mitigation strategies to reduce harm and injury from error [9].

Use of HIT in all pharmacy practice settings is common to the pharmacist’s daily work. Many technologies are integrated into the medication use processes of preparation, dispensing, administering and/or managing the myriad of medications used. Contemporary practice demands that pharmacists master a range of new skills and knowledge in areas of HIT (e.g., e-prescribing, computerized physician order entry (CPOE), health information record interoperability, personal health records, bar coding and robotic medication management), along with systems management (e.g., technology networking, regional and local health information organization link-up, tele-pharmacy) and behavioral care (e.g., medication therapy management services, adherence counseling, disease state management) [8,9,10]. Pharmacists must be able to identify the likely causes of errors with each new technology used to propose local system solutions and strategies to reduce preventable errors [1]. However, there is currently no place where users of HIT can share their risk experiences with HIT products. Further, HIT vendors frequently prohibit the sharing of screen shots (e.g., electronic health records) and technology-specific performance information to further understand patient safety events [11]. Therefore, researchers have a critical opportunity and responsibility to conduct inquiry about the safety events that occur with these technologies and publish these findings for the public good. While some of the safety literature is relevant to pharmacists and pharmacy practice (e.g., many lessons learned from CPOE can also apply to outpatient e-prescribing systems), there is little in the literature examining these problems from a pharmacy-centric framework [1]. There is a major knowledge gap in understanding the risks of harm and injury after specific technologies are integrated into pharmacy practice. The purpose of this study was to understand the impact of specific HITs on patient safety across pharmacy practice settings from the viewpoint of pharmacists in their daily work. This study aimed to: (1) describe the proportion of pharmacists that experience errors or near misses in daily practice, (2) describe the types of error reduction and error occurrence observed by pharmacists related to use of 13 commonly used technologies in daily practice, and (3) describe the error reporting behaviors of pharmacists in practice.

## 2. Materials and Methods

A cross-sectional survey of licensed pharmacists actively practicing across the state of Nebraska gathered both qualitative and quantitative data using a mixed methods approach [12]. The survey was developed to explore the different types of errors and the patient safety impact pharmacists encountered in practice related to the HIT being incorporated into daily work. Pharmacists in Nebraska contributed their knowledge to help identify the patient safety risks and improvements. Pharmacists were asked about observing errors, experiencing errors, error reporting, safety error risk reduction and demographic information including their work environment conditions.

### 2.1. Survey Development

The survey was developed based on a comprehensive literature review, particularly the gray literature related to HIT. The final survey consisted of 22 items in 3 sections: HIT safety impact observations, errors or near misses in daily practice, and demographics about pharmacists’ practice.

A central quantitative question was posed: “Have you been involved in or observed any errors or “near misses” in the last 6 months? (An error is an action which is inaccurate or incorrect, and a near miss is an error you or someone else caught before it reached the patient.)” The six-month time frame was chosen to provide enough of a time window to capture whether an error event occurred, while at the same time short enough that the person’s event memory was likely accurate [13]. Quantitative data was collected using questions designed to solicit closed-ended responses of a dichotomous and numeric response type. Additional areas explored included error reporting and employment conditions. Demographics about the pharmacist respondents were collected. These included pharmacy position, primary pharmacy practice setting, if the respondent worked in additional practice settings, years in practice, hours/week worked, and personal characteristics of age, gender, and race/ethnicity. 

A central qualitative question was posed: “New technology may either reduce errors or increase errors. For the technologies you have used, describe their impact on errors. If you have not used it, leave it blank.” This was followed by the listing of 13 technologies and response areas for open-ended text descriptions labeled as “errors reduced from use of this technology” and “errors observed from use of this technology”. The specific technologies studied included: (a) e-prescribing and CPOE, (b) electronic drug information sources, and clinical decision support rules in the computer system, (c) automated dispensing machines, prescription vending machines, PYXIS system, and Baker cells, (d) bar code scanning and radiofrequency identification tags, and (e) infusion pumps, smart infusion pumps, and insulin infusion pumps. Technologies were identified based upon being widely used in community and/or institutional pharmacy practice and identified in the *Preventing Medication Errors* report [9]. While technologies may overlap in functional category (e.g., automated dispensing machine and Baker cells) the technology was listed on the survey based upon the popular or typical term used by practicing pharmacists in their respective settings. Commonly-used names were included as pharmacists in community practice commonly refer to Baker cells, while pharmacists in institutional practice are likely to refer to automated dispensing machines or PYXIS, for example. Pharmacists were invited to comment on any other miscellaneous technologies they chose to mention. This was followed by an open-ended response question: “Briefly describe a recent story a patient shared with you about a safety concern they had.” These open-ended qualitative responses provided further context and meaning for safety episodes described. 

The face and content validity were assessed by pilot testing the survey with five pharmacists from various practice settings. Modifications were made to the format and content in response to the pilot. The study with survey was reviewed by the Creighton University IRB and classified as exempt IRB Number: 08-14972.

### 2.2. Survey Distribution and Administration

The cross-sectional paper-based survey, *Pharmacists for Patient Safety*, was distributed by U.S. mail to pharmacists in Nebraska between June and August of 2008 using a modified Dillman technique [14]. An invitation cover letter and survey was sent with a business reply return envelope included, followed by a reminder postcard. This process was repeated for three cycles total, with the subsequent cycles being mailed to only those individuals who did not respond. This mode of administration was chosen because many pharmacists do not have access to email in their employment sites, and we did not have email contacts for many of the licensed pharmacists. Using U.S. mail provided us with the opportunity to reach all active licensed pharmacists. The state of Nebraska licensure database was used to identify all 2195 pharmacists who had an active license to practice in the state of Nebraska [15]. The survey had a note informing the respondent about the confidentiality of their response and asking them to respond from the point of view of their primary pharmacy practice location.

### 2.3. Data Analysis

Pharmacists’ quantitative responses to closed-ended survey questions and in vivo qualitative responses to open-ended survey questions were entered into Statistical Package for the Social Sciences (SPSS) to create a comprehensive structured database called the Dyke Anderson Patient Safety Database (DAPSD) [16,17]. The data was quality checked and cleansed to clarify ambiguous and multiple responses. Open-ended responses to the general error and near miss questions were coded and categorized according to the concepts the pharmacist reported about. These were then discussed by the research team members to identify emerging themes interpreted to reflect the prominent points made by the pharmacists. Qualitative in vivo data was coded following text analysis procedures to categorize the various error types reported. These data were further sorted by specific HIT into subcategories to group and relate errors of similar/same type. These data were then transformed to quantitative data to quantify error data categories. Descriptive and relational statistics were performed on all quantitative data. A triangulation design-convergence model guided analysis and interpretation [18].

## 3. Results

### 3.1. Demographics

There were 535 actively practicing pharmacists (24.4%) who responded to the survey. Most respondents were staff pharmacists (71%) from all health care practice settings, followed by owner/managers of community based pharmacies (20%), directors of pharmacy in hospitals/health systems (9%) and a small proportion reporting multiple settings for employment (3%). A small proportion of responding pharmacists (7%) described occupying varied roles including clinical coordinators, supervisors, and relief work pharmacists. Respondents replied from all types of pharmacy settings. Independent pharmacy had the largest proportion of respondents (23.4%), followed by traditional chain (22.4%), hospital (20.4%), grocery chain (12.5%), long-term care (4.7%), ambulatory clinic (2.6%), home care (1.3%) and other (11.4%). Respondents in the “other” category represented pharmacies that specialized in services such as compounding, mail-order, nuclear, home infusion, hospice care, pharmacy benefits management, clinical research and government-affiliated pharmacies within the identified practice settings. One of four pharmacists (23.6%) reported working in at least two locations. The average pharmacist reported working 37 h/week at their primary practice location ranging from 2 h to 87 h/week. Work hour categories revealed that 9% reported working < 20 h per week, 20% between 20–32 h per week, 42% between > 32–40 h per week, 16% between > 40–45 h per week, and 13% > 45 h per week.

There were 302 female pharmacists (56.4%) compared to 233 (43.6%) male pharmacist respondents. The average age of the practicing pharmacist was 46 years (range of 25–91 years). Most respondents were White/Caucasian (95.4%), followed by Asian/Pacific Islander (1.3%), Hispanic (0.7%), and African American (0.4%), with 2.2% declining to indicate their race or ethnicity. These respondents had been licensed pharmacists for an average of 20.9 years, ranging from being in practice less than one year up to 69 years.

### 3.2. Pharmacists Reported Involvement with Errors

One-third of pharmacists reported having been involved in one or more errors sometime in the six months preceding the survey. These errors were of any type, not only HIT-related. An error or a “near miss” was observed by about half of all pharmacists in the last six months. Pharmacists described their own safety stories, with 362 different descriptions of errors or near misses in the previous six months. Pharmacists made it a point to note that errors and near misses are a standard occurrence, even with double checks. It is apparent that pharmacists face patient safety issues frequently every day. These errors and near misses primarily involved incorrect medications, doses, instructions, quantities, formulations, or the wrong patient receiving the medication. Most of the errors did not result in permanent patient harm, but they did cause patients and providers to be inconvenienced, and in many cases, reduced the level of trust the patient previously had with the pharmacist/pharmacy. This representative quote captures many of the stories reported on the survey by pharmacists:
“*Every day we catch errors. I cannot remember any from the past 6 months that have progressed on to the patient. [This is because] our workload is now manageable. Most of the errors I remember were under horrendous workloads.*”

Most of these problems were attributed to workload issues, causing pharmacists to rush and not properly check the medications being dispensed. Only 44% of pharmacists agreed or strongly agreed that their workload does not compromise patient safety, with 24% responding that their workload negatively impacts patient safety. Some pharmacists (16%) felt they do not have enough time to repeat orders on the phone to verify accuracy, only 70% of pharmacists across all practice settings felt that they had the time to perform this action. This is a larger problem in the outpatient setting, where 15% of respondents felt they did not have time to verify verbal orders compared to 5% of their inpatient counterparts. This may also be due to pharmacists in the inpatient setting taking fewer verbal orders overall.

### 3.3. Technology and Safety in Practice

There were 3252 responses about the types of errors that were associated with the 13 different types of HIT studied. These are reports about either error types being reduced or eliminated by integration of HIT (*n* = 1908) or occurring in association with use of the specific HIT (*n* = 1344). Table 1 provides a descriptive breakdown of the number and most common descriptions of these error types characterizing the reports specific to each HIT studied. In all cases, a substantial number of error types were reported. Pharmacists recognized many errors and some error types were eliminated or reduced with the introduction of each HIT. However, pharmacists also reported a large volume of error type observations associated with the integration of HIT. Both the descriptions and the large presence of errors after HIT integration reveals a stark display of risk events commonly encountered in pharmacy practice settings. HIT introduction does not necessarily reduce risk, and changes the types of risks often observed in comparison to when the technology was not in use.

### 3.4. Voluntary Error Reporting by Pharmacists

Overall, 34% of pharmacists in both inpatient and outpatient settings reported that management provides incentives to report errors in their pharmacy. More inpatient practice pharmacists (51%) choose to report errors to external voluntary error reporting programs, compared to 24% of outpatient pharmacists. The focus placed on improving patient safety in the inpatient setting (e.g., The Joint Commission), may be a reason for this disparity. There was a strong correlation between management providing positive incentives for error reporting and the actual reporting of errors to external agencies (r = 0.736; *p* < 0.0001).

## 4. Discussion

Both elimination of error types and creation of new error types is observed with each type of HIT examined in this study. Study findings reveal the frequent, common and integral nature of the errors confronting pharmacists in daily work as technologies intended to reduce risks are incorporated into practice. The nature of technology-specific errors revealed from these reports reinforces the importance of stimulating ongoing error reporting to analyze and design solutions as each of these new error types emerges. Designing safety solutions needs to be a dynamic on-going activity in daily practice. Each specific HIT revealed its own “handprint” of risks that are important to recognize when the technology is used. Pharmacists need to have technology-specific knowledge to design effective solutions to achieve risk reduction and harm reduction for each technology once it is incorporated for use in practice. Each HIT has inherent risks depending upon its functionality and clinical microsystem integration within a pharmacy. Important work is done to reduce risks with technology design pre-marketing by such organizations as the Institute for Safe Medication Practices (ISMP) technology consulting services and the ECRI Institute, and post-marketing voluntary reporting of technology-related safety risks [19,20]. However, identification of safety efforts with specific HIT use is an area that needs further knowledge for local pharmacy practice to achieve risk reduction. 

A multi-stakeholder collaborative report entitled, *Partnership for Health IT Patient Safety*, was released in late 2018 that promotes three safety practice recommendations: (1) identify ways to integrate health IT safety into existing safety programs, (2) convene the necessary stakeholders, including users, vendors, organizations and patients to actively collaborate on safety, and (3) embed safety into the culture and daily workflow to achieve a unified vision of health IT safety [20]. While this report is not focused on medication-related HIT, its creation points out concerns about HIT safety overarching healthcare today. A highlight of this report is about safety improvement needs with electronic health records (EHRs). The report highlights the case that has been developed to study EHR safety from the unique aspects of each product and the clinical microsystem it is embedded within. A safety assessment model has been adopted by the Office of the National Coordinator for Health Information Technology (ONCHIT) to be applied by health care delivery professionals within care systems to simultaneously learn how to improve the EHR product and systems surrounding its use. This model, produced as the Safety Assurance Factors of EHR Resilience (SAFER) Guides, is designed to help healthcare organizations conduct self-assessments to optimize the safety and safe use of EHRs using a checklist system of recommended practices. Each technology in use should have a similar commitment to risk reduction in our health care delivery systems; in our pharmacies [21]. 

Collecting and reporting pharmacists’ experiences with HIT is essential to the goal of improving products and their use across healthcare delivery to promote patient safety. It is common for health care providers to experience frustration when transitioning to “better” technologies, resulting in many “opting out” of use when such choices are voluntary. But much of HIT use is no longer voluntary. Even after new HIT is in place, issues such as lack of interoperability between systems, devices, and humans are very common. For example, e-prescribing is dominant nationwide, estimated to be integrated into 98% of pharmacies and used by 69% of prescribers [22]. While e-prescribing may save pharmacists time by reducing handwriting issues, it also can require more time to clarify a prescription if the physician inadvertently selects the wrong medication through a drop-down menu error [2]. This example demonstrates the suboptimal realization of the intended value of many technologies. The “theoretical’ benefits of HIT implementation into practice may not always be fully realized or are realized slower than expected; and the roller coaster of HIT integration has no end in sight.

Pharmacists are called upon to play a key and central role in patient safety across all practice settings, leveraging HIT for safety and improved care. Our study’s findings imply that pharmacists’ fears of being involved in the inevitable error that hurts someone and patients’ fears about a possible injury or harm and health are well founded. Research with consumers who use community pharmacy services reveals that patients are more likely to report medication errors to a pharmacist than a physician, and they view the pharmacist as the final interceptors to detect medication errors before reaching themselves [23]. Pharmacists are viewed as the gatekeepers of safety.

This has important implications to pharmacy practice. Fundamentally, the human-technology interactions between pharmacists, patients and HIT use are integral to assure the safe and effective medication use system. Technology does not in and of itself provide the necessary support of patient care without the stewardship of the professional expert pharmacist. Findings from this work suggest that the professional pharmacist is essential to assuring the patient’s safety by both assuring the correctness of the technical aspects of care delivery and communicating the specifics of safe medication use to prescribers and patients. The pharmacist is the gatekeeper in the medication use system. While health care efficiency constantly pushes on eliminating “expenses” to be affordable, this work clearly indicates that errors leading to unsafe events would be frequent and unchecked without the expertise of the pharmacist. To achieve on-going safety for patients, pharmacists must be vigilant about their expertise and competence with these technologies. 

Pharmacist reporting is an essential activity for the purposes of learning and improving the patient safety experience for patients. On-going educational efforts that emphasize safe practices and safe decisions specific to each HIT is an essential area where the profession must structure an on-going risk discover model, development of solutions strategies and self-educate. Organizations should put incentives in place that will support pharmacist reporting of patient safety problems associated with HIT use. Failure to properly support reporting behaviors will lead to the persistence of HIT-related error types; growing in magnitude as these technologies continue to grow in use.

### Limitations

Only 25% of pharmacists responded to the survey. However, we did receive the observations of 535 pharmacists from the entire licensed and active pharmacist workforce in the state. Having one quarter of the workforce respond is a wealth of information about HIT use and safety. The primary concern with the lower response rate is the increased likelihood of non-response bias. It is possible that the breadth of error reports described in this study by the respondents is under or overreported compared to the total number of events during that same time. Under-reporting is highly likely because of employment non-disclosure guidelines that many employers have requiring that a pharmacist not discuss patient safety events with anyone. If we were trying to determine an error rate, this would be of great concern. However, we were not looking to determine an error rate. Instead we were operating under the framework that if an error occurs once then the opportunity exists for it to happen again. We were interested in a rich description of the types of safety events that were experienced by pharmacists who worked with these technologies daily, not a precise measure of their rate of occurrence. Identification and rich description of these safety events provides us with foundational knowledge that can be used to better understand how to maximize safety improvements while addressing new and/or ongoing safety concerns. As such, these findings provide a meaningful description of both safety improvements and unsafe events associated with specific technologies.

With many products, a competitive marketplace would result in substantial changes in response to demand for technology modification to achieve performance improvements. This would suggest that our study of technologies might be outdated since its completion was over ten years ago in 2008. However, changes in healthcare delivery, use of HIT and the nature of errors impacting patient safety in the past decade have been slow [24]. HIT products have not changed considerably or advanced adequately, referred to as a market failure, because there has been little opportunity for cross-company product comparisons due to industry proprietary restrictions on information sharing about product performance and safety risk [25]. This is illustrated by two cases: infusion pump safety problems and ADMs.

From 2005 to 2009, the FDA received more than 56,000 Medical Device Reports (MDRs) about infusion pumps with an increase in the number of reports in 2010 [26]. This led the FDA to start the Infusion Pump Improvement Initiative in 2010, to leverage infusion pump manufacturers to improve the safety performance of these devices, as they were not doing so on their own. Today, the FDA program remains highly active and vigilant about these devices and the manufacturers, continuing to solicit and obtain MDRs from health professionals based upon the program website update in 2017 [26]. While it is clear that patient safety problems continue, the FDA has not provided a quantitative summary of the frequency of reports since 2010.

Similarly, ADMs are in wide use, with 97% of US hospital pharmacies using them in some capacity as of 2014, and 70.2% of U.S. hospital pharmacies indicating ADMs as the primary method of maintenance dose distribution in 2017 [27,28]. Despite this adoption, the literature still raises concerns about patient safety issues with ADMs, noted by this literature from the nursing professions point of view [29]. Unfortunately, little is available in the published literature that documents the safety concerns and observations about unsafe ADM equipment and use given the integration of this technology nationwide in pharmacy practice settings. It is not the case that there are no problems, rather, it is the case that we are not publishing about these problems.

So, what is being reported about? The evidence reports are focused on the EHR. This critical HIT has drawn almost all of the attention and resources to study safety problems. So much so, that a contemporary assessment published in 2018 by top thought leaders Bates and Singh, suggests emerging priorities beyond the record itself [30]. They remind us that while HIT can help prevent many types of patient safety errors, HIT also continues to introduce new problems. They identify a key emerging priority for patient safety in 2018 is ensuring the safety of the technology itself, its safe use, and the effective use of it to improve patient safety. In terms of this study, a more recent survey might now include pharmacists with more experience with HIT and different perspectives, even if errors haven’t changed. Overall, the findings from our study remain meaningful as the results raise our awareness and continue to create the opportunity for safety improvement with specific HIT used in pharmacy practice.

## 5. Conclusions

The introduction of HIT is often assumed to reduce harm and injury. This belief is embraced so fully in the U.S. that federal agencies have catalyzed the development of HIT and supported its adoption and use to measurably reduce errors. However, to date, the health care industry has not adequately kept standard setting and performance toward safer practices to the level of scientific inquiry required for each unique technology introduced. This work reveals a need to further study individual HIT on an ongoing basis, to obtain a differential profile of the safety events likely to occur with that technology. While a survey conducted today may glean responses representing the perspectives from more HIT-experienced pharmacists, the safety concern has not changed. Data is needed for pharmacist training to ensure that each new technology can be operated in a way to maximize its safe use. To achieve this, ongoing local reporting and subsequent analysis of root causes of errors or safety events need to be pharmacist-led at the local level. Results should drive ongoing education of safety risks and best practices emphasizing risk reduction within the pharmacies where such technologies are integrated for use.

Integration of new technologies into pharmacy practice introduces new error types that have a substantial presence in the pharmacist’s daily work. Each specific HIT needs to be studied for safety problem types and ongoing education focused on reducing the risks of each technology integrated into the pharmacist’s professional on-going education. Both the profession of pharmacy and each individual pharmacist must adopt an ongoing continuous improvement method at the local level of practice to minimize risks to patients and optimize health care outcomes.

## Figures and Tables

**Table 1 pharmacy-07-00007-t001:** Number of pharmacists reports of types of errors reduced or observed after integration of specific technologies.

Technology	Types of Errors Reduced After Integration of Specific Technology		Types of Errors Observed After Integration of Specific Technology	
	*n*	Most Common Descriptions (*n*) ^a^	*n*	Most Common Descriptions (*n*) ^a^
Electronic prescribing	234	Legibility errors eliminated. (170) More accurate and complete information provided. (21) Time savings resulting from not needing to clarify information. (11)	237	Wrong information provided. (129) System information missing or incorrect, i.e., drug product information. (51) ○ “1 ½ displayed as 1—system doesn’t recognize fractions.” ○ “system does not receive information correctly-directions may get messed up.” ○ “Doctor ordered wrong drug strength and our system NDC does not match doctor’s NDC. We must retype drug by memory.” Drop down menu selection error. (46) ○ “Wrong drug selected from.” ○ “Quantity may not match day supply calculated by s.i.g.”
Computerized prescriber order entry	201	Legibility problems. (118) ○ “Legibility corrected.” ○ “Decimal placing is clear.” Faster; saves time. (6)	150	Wrong information: drug, dose, patient. (114) ○ “Often wrong s.i.g. or wrong drug form (XL, SR, or LA) are sent across.” System incompatibilities—drug not in system. (21) ○ “Confusing format with additional directions in a different field than original directions.” ○ “Some systems put in default directions.” ○ “Physician writes another set of directions, both are sent—we have to clarify which is correct.”
Clinical decision support rules in computer system	108	Decrease in interactions—allergies, duplications, diseases, drug-drug. (36) ○ “Drug/disease interactions caught.” ○ “Allergy checks.” ○ “Duplicate therapy identified.” ○ “Drug Use Review intervention notices on the computer.” Provides double-check for dose. (10) ○ “Proper record dosing for drug alerts about proper dosage form.”	69	Pharmacist experiences alert fatigue/overrides/ignore warnings. (22) ○ “Too many irrelevant warnings can cause inattention to this system.” ○ “Missed interactions because of not fully paying attention to override.” ○ “Alert Fatigue—makes you ignore alerts.”
Electronic drug information sources	205	Increased accessibility—faster and/or easier to find information. (95) ○ “Quick and easy access to information- confirm counseling—usually have time to do this now when compared to getting information from a book previously.” ○ “Easier to find information instead of searching package insert.” More current/complete information on drug dosage, drug interactions. (92) ○ “Up-to-date information for new dosing or products.” ○ “More detailed info readily available. Fewer problems with correct dosages, drug interactions and IV compatibilities.” ○ “Increased consistency of information/dosing” ○ “Appropriateness of therapy, dosing, administration, multiple clinical applications”	114	Not enough information. (15) ○ “Need to have more. OTC, herbal, and vitamin information on-line.” ○ “May not include all information needed.” Computer/system failure. (13) ○ “Cannot be accessed if computer malfunctions or link is down.” Difficulty in interpreting information. (13) ○ “RNs have misinterpreted IV compatibilities by not looking at diluents and concentrations.” ○ “Incorrect interpretation of information leading to errors.” Information is not current. (9) ○ “Relying on information as up to date.” ○ “Not always up to date e.g., black box warning.” Too much information. (7) ○ “There is too much information—there needs to be summaries.” Reliability of sources. (5) ○ “Need to watch source of information to make sure it’s reliable source.” Inaccurate information. (5) ○ “Errors in information.”
Automated dispensing machines	220	Increased accuracy—correct drug, dosage, number of tablets/capsules. (133) ○ “Accurate acquisition of drug and count.” Saves time. (24) ○ “More time to spend on counseling and verification.” Better regulation of controlled substances. (9) ○ “Better accountability for narcotic use.” “Robotic dispensing: labels on perfectly, decreased wrong drug and decreased wrong quantity.”	178	Incorrect drug loaded into machine. (71) ○ “Human error—wrong drug dispensed due to incorrect use or filling of machines.” Incorrect counting—often due to broken tablets. (42) ○ “Wrong number of pills counted due to static or tablet weight.” ○ “Broken tablets cause inaccurate counts.” Broken tablets. (15) Override via nurses. (7) ○ “RN’s may remove drugs on override and can choose wrong drug or dosage form.”
Prescription vending machine	78	Right drug selected. (7) ○ “Accurate drug acquisition, and count.” ○ “Selection of wrong product reduced.” Accurate counting. (5)	49	Incorrect drug loaded into machine. (7) ○ “Incorrect stocking” Incorrect counting. (2) ○ “Counting errors (if the cassette runs out, but the robot doesn’t know).”
PYXIS system	142	Accurate drug/dosage/patient. (44) ○ “Wrong medicine, wrong dose, wrong patient, wrong counts.” Documentation/inventory control. (21) ○ “Documentation on errors made by nurse.” ○ “Inventory control decreased errors.” “Accountability for narcotics much better to track issues.”	90	Incorrect drug loaded into machine. (37) ○ “Putting incorrect medicines in the wrong slot either by pharmacy tech (error rate lowered now that we have bar codes) or by nurse returning med to drawer.”
Baker cells	110	Increased accuracy—correct drug, dosage, number of tablets/capsules. (46) ○ “Decrease wrong drug. Decrease wrong quantity.” Saves time. (12) ○ “Faster filling.”	88	Incorrect drug loaded into machine. (41) ○ “Wrong drug filled in baker cells—human error.” Miscounts. (22) ○ “When cell runs out wrong quantity can be dispensed.”
Bar code scanning	249	Accurate drug/dosage/patient. (179) ○ “Patients getting correct medications.” ○ “Prevents wrong drug, strength, or dosage form from being dispensed.”	156	Overrides/workarounds. (50) ○ “Barcode scanning done AFTER drug given.” ○ “Only scan 1 bottle but use from 2 stock bottles and 2nd bottle was not the same; or scan 1 bottle but use bottle behind on shelf and bottle behind wasn’t the same product.” System incompatibilities. (34) ○ “Medicines from new manufacturer (different generic drug companies) might not scan.” ○ “If bar code is too small have to type in NDC number.” ○ “If bar code is UPC code, does not co-relate to product NDC—incorrect drug comes up.” ○ “Often does not allow for difference in package sizes even though drug is correct” Mislabeled barcode. (13) ○ “Wrong drug packaged in bar coded bottle.”
Radiofrequency identification tagging	71	“Dispensing/pick errors.” (1) “Right patient.” (1) ○ “Ensures the right patient is getting the drug.” “Pedigree.” (1) “Reduce counterfeit drugs.” (1) “Not used/no experience.” (8)	34	“Failure to recognize ½ tab doses.” (1) “FDA—needs better control—higher involvement better design.” (1) “No access in power outage—must remember physical badge.” (1) “Not used/no experience.” (4)
Infusion pumps	117	Increased accuracy—calculations/ programmed rates. (42) ○ “Less errors by calculations by RNs and pharmacy.” ○ “Rate more consistent and accurate.” ○ “Prevented inadvertent overdose.” ○ “Now have Alaris pumps we programmed w/ “rails.” So, if nurse programs to run outside set rails, pump will not run.”	79	Programming errors. (24) ○ “Programmed incorrectly.” ○ “Can’t change concentrations unless reprogram pump.”
Smart infusion pumps	97	Increased accuracy—calculations/programmed rates. (30) ○ “Calculates complex dose and correct infusion rate.” ○ “Using built-in programs decreases incorrect drip rates.” ○ “Lockouts provide safety features (MAX—MIN dosing and rates).”	56	Programming errors/entering wrong drug or dose. (18) ○ “Programmed wrong or wrong face plate.” ○ “Pumps programmed incorrectly into drug library (ie entered in as mL/Hr vs. mcg/Hr).” ○ “Medicine and concentration must be pre-loaded.” Overrides/workarounds. (7) ○ “Nursing staff over-rides pump settings and gives drug too fast/too slow.”
Insulin infusion pumps	76	Increased glycemic control. (15) ○ “Tighter glycemic control.” ○ “Decrease injections having to be drawn up.” ○ “Missed doses.”	44	Incorrect programming/set up by humans. (6) ○ “Pump can be programmed erroneously by staff.”

^a^ Counts of observations in the “Most Common Descriptions” table columns will not add up to the total “*n*” reports in the relevant “Types of Errors” column. This table only includes the commonly reported observations.

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
