# Peer review of "Health Information Technology Use and Patient Safety: Study of Pharmacists in Nebraska"

_pharmacy, 2019, doi:10.3390/pharmacy7010007_

Round 1
Reviewer 1 Report
1 Overall
This paper describes a study that is interesting from a practical perspective, and is likely to resonate with many readers. A major issue is that the study appears to utilise data that are 10y old – albeit the results seem relevant to current practice. A more recent survey might include pharmacists with more experience with HIT and different perspectives, even if errors haven't changed. What is the reason for the large delay between conducting the survey, and submitting for publication? Specifics follow below.
2 Abstract
· The year the survey was conducted should be stated – I suspect most readers expect to see this in the abstract.
3 Main text
· Introduction
o nil
· Methods
o Survey development:
§ why “in last 6 months”? instead of e.g. 12 months?
§ Was the need for relevance to medicines specified? e.g. not using appropriate infection control techniques when seeing a patient with acute gastroenteritis is an error, which might be propagated by the information about needing to use these techniques when seeing the patient being only in a difficult-to-find place in the electronic health record
§ Please include the actual survey as an appendix. Without this, the study is more difficult to a) judge and b) replicate.
§ How were the 13 technologies identified? There are a number of overlapping technologies – why? Unclear how e-prescribing and CPOE are different. E-prescribing is usually considered to be a subset of CPOE (with other types of CPOE including e.g. radiology and laboratory requests). Similarly, PYXIS is a subset of automated dispensing machines. Given the overlaps, would make sense to combine results for overlapping technologies.
· Results
o Demographics
§ 61% + 20% + 9% is only 90%. Unclear if the 7% with varied roles overlaps with this 90% or not. Even if the 7% are in addition to 90%, still leaves 3% missing.
§ “46 years (25 – 91 years)”: specify what average means - mean/median/mode/geometric mean etc.
o Voluntary error reporting
§ “p < .000”: P cannot be < 0, please correct.
o Table 1
§ The narratives are brief but provide reasonable detail and insight – well done
§ “Alert fatigue”: please indicate who was subjected to this e.g. was this an observation by pharmacists of what prescribers experienced, or did pharmacists experience this themselves, or both?
· Discussion
o “the ongoing continuing incorporation of safety efforts with specific HIT use is not enculturated into local pharmacy practice, unlike the way that integration of medication error risk reduction is. The pharmacy profession must get to that same level of engagement and risk reduction with incorporated technologies.” This is internally inconsistent. In a HIT work environment, medication error risk reduction, by definition, includes safety efforts with HIT use. This has been the case in all institutions that I've worked with that have e-prescribing - considering HIT as part of medication error risk reduction work has been a natural extension to pre-HIT risk reduction work. However, there are HIT-specific barriers to this work with HIT i.e. lack of cooperation from IT services, whether it be the e-prescribing vendor through to local health institution IT support. Is this what the authors mean?
o Limitations: please comment on ~25% response rate. Perhaps only pharmacists who've noticed problems responded? i.e. these data overestimate the true rate of errors? Or pharmacists overwhelmed with errors and who are nihilistic about errors did not respond i.e. these data underestimate the error rate?
the authors need to justify that this is due to ‘incompetence’ (e.g. inadequate resource to collate data), rather than ‘conspiracy’(e.g. undeclared financial conflict of interest). The actual survey questions should also be included - a standard expectation to accompany such manuscripts these days.
Author Response
A file with the response is attached.

Reviewer 2 Report
This was an interesting piece of work. Below are suggestions as to how it could be improved:
Abstract:
Background/aim: Your aim/objectives could be more overt
Methods: Was the survey pre-piloted and ethically approved? Provide brief mention as to what data analysis was undertaken.
Results/aim: It was a bit confusing when you stated that you were seeking to “ascertain specific patient safety issues encountered with different technologies” yet then you reported that errors “were reduced or eliminated by integration of HIT”. That seems more like you were ascertaining the impact of HIT on patient safety, rather than trying to establish what patient safety issues were evident due to its use/adoption. I think you need to broaden your aim/reword it as your work doesn’t appear to be solely about identifying issues/problems.
Results: You state: “Of 535 participants, one third of pharmacists reported having been involved in an error in the six months preceding the survey, and half observed an error or “near miss”. Is this finding directly linked to HIT/to your study aim?
Results/conclusion: Could you mention one or two examples of such errors? It’s hard for readers to appreciate your conclusion about ‘new errors’ when no detail about the errors has been provided.
Main manuscript:
Lines 61-63 You state: “This study serves to educate us all about the nature of specific technologies…” – I would consider removing or rewording this as I am not sure your study did that (possibly it educated on the benefits and risks associated with using the technology in practice). I also think it would be useful to have explicit study aims/objectives.
Lines 74-76 You state: “A central quantitative question was posed: Have you been involved in or observed any errors or “near misses” in the last 6 months?” Was this question directly linked to HIT (since the focus of the work was HIT-related patient safety issues and not just patient safety issues in general)? So xx% errors/near misses in total, of which yy% were deemed to be linked to HIT.
Line 72 (starting point) Survey development – perhaps include the total number of questions and/or sections and what you did to try and maximize the survey response rate from the outset. Did you collect any identifiable demographic data (you have stated ‘these included…’ suggesting other information was sought, so it would be good to know if this encompassed the collection of any identifiable data). You might need to expand on some of the terms, for example ‘PYXIS system’ (and include ™ or the generic description, if applicable).
Line 99 (starting point) Survey administration – you could expand on this section slightly (including the number of opportunities participants had to complete the survey), particularly for readers who are unfamiliar with the modified Dillman technique.
Table 1 suggests you were ascertaining the impact of HIT on patient safety (since you report both positives and negatives), rather than solely “patient safety events and concerns across pharmacy” (which was previously stated on Line 62). I think you need to broaden your aim.
Line 251 Limitations: Should mention your low response rate (24.4%) here and the increased likelihood of non-response bias.
Discussion/conclusion: you could expand on the implications for practice.
Author Response
A file with the response is attached.

Round 2
Reviewer 1 Report
Thanks for the opportunity to comment again.
The authors have improved their manuscript to address many of the minor comments previously raised. However, the following is a copy and paste from the first paragraph of my previous report to the authors, as it remains to be resolved by the authors.
A major issue is that the study appears to utilise data that are 10y old – albeit the results seem relevant to current practice. A more recent survey might include pharmacists with more experience with HIT and different perspectives, even if errors haven't changed. What is the reason for the large delay between conducting the survey, and submitting for publication?
Author Response
Reviewers comments:
Thanks for the opportunity to comment again. The authors have improved their manuscript to address many of the minor comments previously raised. However, the following is a copy and paste from the first paragraph of my previous report to the authors, as it remains to be resolved by the authors.
A major issue is that the study appears to utilise data that are 10y old – albeit the results seem relevant to current practice. A more recent survey might include pharmacists with more experience with HIT and different perspectives, even if errors haven't changed. What is the reason for the large delay between conducting the survey, and submitting for publication?
Author (Galt) response:
Thank you for reasking this point that I did not address. The delay was for personal reasons, not scientific. I faced personal and career challenges that overwhelmed my ability to efficiently complete some of the important work I was involved in, a poignant trade off with other priorities during a several year period. It is should have been submitted much sooner, of course. When I was able to return to focus my attention more fully on my work, I realized that this work was worthwhile to get published for others to use as evidence and support for patient safety claims with HIT. The reviewer is right on that tough call about how long it has taken to get this to an outlet...and also right about its current relevance. I am sorry that I did not move this up the priority list sooner than now as the results could have been useful sooner.
Reviewer 2 Report
Thank you for addressing the majority of my comments and concerns.
I just have a minor comments at this stage:
I would write the aim/purpose in past tense (this study aimed to) rather than present tense (aims to), since the study is now complete/over. This applies to both the abstract and main text.
Author Response
Reviewers comments:
Thank you for addressing the majority of my comments and concerns.
I just have a minor comments at this stage:
I would write the aim/purpose in past tense (this study aimed to) rather than present tense (aims to), since the study is now complete/over. This applies to both the abstract and main text.
Authors response:
Thank you for this recommendation. The abstract was modified to the past tense (see below):
"This study aimed to describe the impact of 13 different health information technologies (HITs) on patient safety across pharmacy practice settings from the viewpoint of the working pharmacist."
The purpose statement within 1. Introduction section was modified to past tense (see below):
"The purpose of this study was to understand the impact of specific HITs on patient safety across pharmacy practice settings from the viewpoint of pharmacists in their daily work. This study aimed to: (1) describe the proportion of pharmacists that experience errors or near misses in daily practice, (2) describe the types of error reduction and error occurrence observed by pharmacists related to use of 13 commonly used technologies in daily practice, and (3) describe the error reporting behavior of pharmacists in practice."